# A Mini-Review on Solid Lipid Nanoparticles and Nanostructured Lipid Carriers: Topical Delivery of Phytochemicals for the Treatment of *Acne Vulgaris*

**DOI:** 10.3390/molecules27113460

**Published:** 2022-05-27

**Authors:** Romchat Chutoprapat, Peerawas Kopongpanich, Lai Wah Chan

**Affiliations:** 1Department of Pharmaceutics and Industrial Pharmacy, Faculty of Pharmaceutical Sciences, Chulalongkorn University, Bangkok 10330, Thailand; peerawas.k@student.chula.ac.th; 2Department of Pharmacy, Faculty of Science, National University of Singapore, Singapore 117543, Singapore; phaclw@nus.edu.sg

**Keywords:** solid lipid nanoparticles, nanostructured lipid carriers, topical application, phytochemicals, *acne vulgaris*

## Abstract

*Acne vulgaris* (acne) is one of the most common dermatological problems affecting adolescents and young adults. Although acne may not lead to serious medical complications, its psychosocial effects are tremendous and scientifically proven. The first-line treatment for acne is topical medications composed of synthetic compounds, which usually cause skin irritation, dryness and itch. Therefore, naturally occurring constituents from plants (phytochemicals), which are generally regarded as safe, have received much attention as an alternative source of treatment. However, the degradation of phytochemicals under high temperature, light and oxygen, and their poor penetration across the skin barrier limit their application in dermatology. Encapsulation in lipid nanoparticles is one of the strategies commonly used to deliver drugs and phytochemicals because it allows appropriate concentrations of these substances to be delivered to the site of action with minimal side effects. Solid lipid nanoparticles (SLNs) and nanostructured lipid carriers (NLCs) are promising delivery systems developed from the combination of lipid and emulsifier. They have numerous advantages that include biocompatibility and biodegradability of lipid materials, enhancement of drug solubility and stability, ease of modulation of drug release, ease of scale-up, feasibility of incorporation of both hydrophilic and lipophilic drugs and occlusive moisturization, which make them very attractive carriers for delivery of bioactive compounds for treating skin ailments such as acne. In this review, the concepts of SLNs and NLCs, methods of preparation, characterization, and their application in the encapsulation of anti-acne phytochemicals will be discussed.

## 1. Introduction

The cause of acne is multifactorial, with four primary factors including excess sebum production, *Cutibacterium acnes* (formerly *Propionibacterium acnes* (*P. acnes*)) colonization, follicular hyperkeratinization, and release of inflammatory mediators into the skin. There are standard acne treatments that vary according to the stage or severity of the disease. The first-line treatment for mild to moderate *acne vulgaris* is topical medications composed of synthetic compounds or their combination with oral antibiotics. However, most of these topical medications usually cause side effects such as irritation, dryness, scaling and itch to the skin. Therefore, the use of phytochemicals has been studied as an alternative treatment to resolve unpleasant side effects due to the synthetic compounds. For example, α- and γ-mangostins, the active compounds in mangosteen, were able to reduce the proliferation of keratinocytes induced by *P. acnes* and suppress *P. acnes*-induced inflammation [1]. In another study, eucalyptus oil was found to decrease sebum production in a rat sebaceous gland model and inhibit secondary infection caused by other strains of bacteria [2]. Resveratrol, a natural compound produced in some fruits such as grapes, has been shown to be effective in reducing the number of acne lesions in a clinical study [3]. Nevertheless, the major challenges of phytochemicals derived from the plants are their instability and poor penetration across the skin barrier which may limit their application in dermatology. The use of lipid carriers, such as solid lipid nanoparticles (SLNs) and nanostructured lipid carriers (NLCs), can increase the absorption rate and facilitate the sustained release of active substances. Moreover, these carriers can protect the active substances from degradation [4]. Solid lipid nanoparticles (SLNs) are nanospheres prepared from solid lipids and surfactants. SLNs have been applied in a wide variety of cosmetics and pharmaceutical preparations due to their various advantages such as good stability, rigid morphology, good biocompatibility, ease of scale-up, ease of modulation of drug release, and the avoidance of organic solvents in the preparation [5,6]. Moreover, SLNs have an occlusive property which can reduce the trans-epidermal water loss and make the skin hydrated [7]. However, SLNs show a low drug loading capacity. Therefore, nanostructured lipid carriers (NLCs), the second generation of lipid nanoparticles, were developed to address this problem of SLNs. NLCs are prepared by incorporating liquid lipid to solid lipid in order to impart imperfections to the crystal order of the solid lipid, thereby facilitating the incorporation of higher amount of drug while preserving the stability of the NLCs [6]. Both SLNs and NLCs can play an important role in anti-acne therapy owing to their ability to penetrate into the hair follicle and sebaceous gland, occlusive and skin hydration effects [8].

The focus of this review work is to provide a concise overview of solid lipid nanoparticles (SLNs) and nanostructured lipid carriers (NLCs) as promising carrier systems for topical delivery and to highlight the application of SLNs and NLCs in the encapsulation of anti-acne phytochemicals. The concepts of SLNs and NLCs, their recent application in the encapsulation of anti-acne phytochemicals as well as methods of preparation and characterization have been reviewed and discussed below.

## 2. Solid Lipid Nanoparticles (SLNs)

SLNs were introduced in 1990s as an alternative to conventional colloidal carriers such as liposomes and polymeric nanoparticles due to the possibility of production at large industrial scale [9]. Moreover, SLNs have gained much interest in pharmaceutical sciences for drug delivery applications because of their flexibility in modulating the drug release, high drug loading capacity, and protecting the drugs from physical and chemical degradation [10]. The SLNs are nanosized lipid particles comprising solid lipids dispersed in aqueous surfactant media. The lipids commonly used in the preparation of SLNs are biodegradable lipids with high melting point such as triglycerides, partial glycerides, fatty acids, fatty alcohols, and waxes [11]. Surfactants are used to stabilize the structure of SLNs by decreasing surface tension between aqueous media and lipid [12]. It was reported that the higher the surfactant concentration, the smaller the particle size of SLNs obtained. However, a high concentration of surfactants may cause toxicity [13]. Therefore, the surfactant concentration used in the development of SLNs has to be optimized. Particle size of SLNs is one of the most important parameters affecting the drug delivery efficiency. Some studies have shown that the skin permeability of SLNs increases when their particle size decreases, and the sub-100 nm size range is optimal for skin delivery because they can penetrate into deeper skin layers through hair follicles [14]. In addition, SLNs may enhance the penetration of drugs through the stratum corneum due to their occlusive property [7]. The incorporation of drugs into SLNs can be described by three models, namely homogeneous matrix, drug-enriched shell, and drug-enriched core (Figure 1).

Homogeneous matrix model: the drug is dispersed in lipid matrix without the use of solubilizers or surfactants. This model is usually prepared by the cold homogenization technique [15].The drug-enriched shell model: a mixture of lipid and drug is heated at temperature above the melting point of the lipid. On rapid cooling, lipid precipitates at the core whereas the drug is concentrated at the outer melted lipid. The drug-enriched shell is completely formed when the melted mixture is cooled to room temperature [16].The drug-enriched core model: the concentration of drug in the melted lipid is close to its saturation solubility. The cooling process creates supersaturation of the drug in the melted lipid, resulting in drug precipitation at the core prior to lipid crystallization. Further cooling will lead to the crystallization of the lipid surrounding the drug core as a shell [17]. The drug-enriched shell and the drug-enriched core models are usually produced by hot homogenization technique.

**Figure 1 molecules-27-03460-f001:**
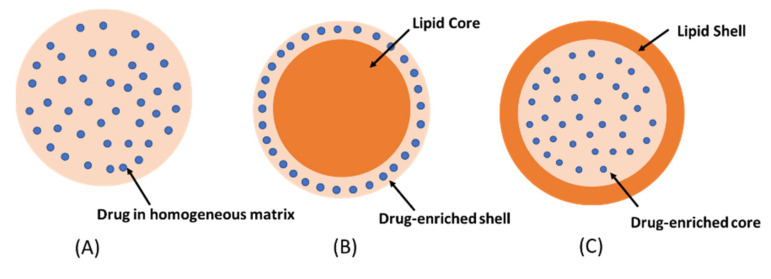
The drug incorporation models of SLNs (**A**) homogeneous matrix, (**B**) drug-enriched shell, (**C**) drug-enriched core.

Advantages of SLNs [18] are:SLNs are biodegradable and biocompatible.They can modulate the drug release.SLNs can enhance the penetration of drugs through the stratum corneum.They have occlusive property which can increase skin hydration.SLNs can be sterilized using the autoclave. The high temperature (121 °C) during sterilization by autoclaving certainly causes a hot o/w microemulsion. On subsequent cooling, the SLNs reformed. However, it should be noted that the average particle size of SLNs is usually increased after sterilization by heating.Ease of large-scale production and low production cost.They are safe because of avoidance of organic solvents in the preparation.Incorporation of lipophilic and hydrophilic drugs is feasible.They can increase storage stability of loaded drug.


Limitations of SLNs [7,19] are:
Low loading capacity and drug expulsion during storage. Since the lipids crystallize in high energy modification (α form) during the preparation, they can transform to more organized, lower energy modification (β form) during storage. This modification resulted in fewer imperfections of lipid matrix for drug loading leading to drug expulsion.The irreversible transformation of a low viscous SLNs dispersion into a viscous gel, known as gelation phenomena, may occur rapidly and is unpredictable upon storage. In this circumstance, the surfactants can no longer stabilize the new surfaces, and hence the particle aggregation occurred. It has been reported that the addition of co-emulsifying surfactants with high mobility such as glycocholate can retard a gelation in SLNs.Irritation and sensitizing potential of some surfactants. Three surfactant types including cationic (such as cetylpyridinium chloride, cetyltrimethyl ammonium bromide), anionic (such as sodium dodecyl sulfate, sodium glycocholate), and non-ionic (such as poloxamer, Tween, phospholipid, Cremophor) are generally used in the preparation of both SLNs and NLCs. They are chosen based on their hydrophile-lipophile balance (HLB) values, effects on the particle size and lipid modification of the lipid carriers, as well as the route of administration of the lipid carriers [20]. It has been reported that the type and concentration of surfactants exert influence on the potential toxicity of these lipid carriers. According to the studies of Scholer et al. (2001), SLN formulation containing cetylpyridinium chloride exhibited strong cytotoxic effect on murine peritoneal macrophages, whereas other SLN formulations containing poloxamer, Tween, phospholipid, and sodium dodecyl sulfate at the same concentration, reduced cell viability slightly [21]. Furthermore, cationic and anionic surfactants are broadly regarded as potent irritants to human skin, with cationic surfactants being more cytotoxic than anionic surfactants. In contrast, non-ionic surfactants are considered to be safe (lowest irritation potential) [22]. Therefore, the safety test of SLNs and NLCs should be investigated prior to in vivo use of these lipid carriers.


## 3. Nanostructured Lipid Carriers (NLCs)

Nanostructured lipids carriers (NLCs) were developed to improve encapsulation capacity and storage stability of SLNs. The NLCs are composed of a mixture of solid and liquid lipids, with typical percentage of liquid lipid in the range of 10*–*30%. The presence of liquid lipid in the lipid mixture produces a nanostructure matrix which can accommodate a greater load of drug [23]. In addition, the high loading capacity of NLCs will enable a high drug concentration gradient on the skin, which will in turn improve drug permeation [7]. NLCs also have various advantages similar to SLNs such as the use of biodegradable and biocompatible lipids, controlled drug release, enhanced drug penetration and stability, ease of fabrication and avoidance of organic solvents in preparation [24]. Based on the nature of lipid content and ratios of solid and liquid lipids, NLCs can be classified into three types including imperfect crystal type, amorphous type, and multiple oil-in-fat-in-water (O/F/W) type (Figure 2) [25].

Imperfect crystal type: this type involves mixing of spatially different liquid lipids such as glycerides and solid lipids which introduce imperfections in the crystal order leading to more space for drug loading. The imperfection can be increased by using a mixture of various glycerides which vary in saturation and length of carbon chains.Amorphous type: this type is formed by incorporating special liquid oils such as isopropyl myristate or hydroxyoctacosanyl hydroxystearate in a lipid matrix. The matrix will solidify in an amorphous form that potentially reduces the expulsion of the loaded drug by delaying the crystallization of lipids during the preparation and storage of the NLCsMultiple O/F/W type: this type is formed by adding high amount of liquid lipid beyond its solubility in the lipid matrix. This will create oil nanocompartments distributed in the solid matrix. Drug solubility in oil nanocompartment is higher than in solid matrix which enables higher drug loading. Moreover, a solid lipid matrix around the oil nanocompartments acts as a barrier that prevents drug leakage and provides controlled drug release.

**Figure 2 molecules-27-03460-f002:**
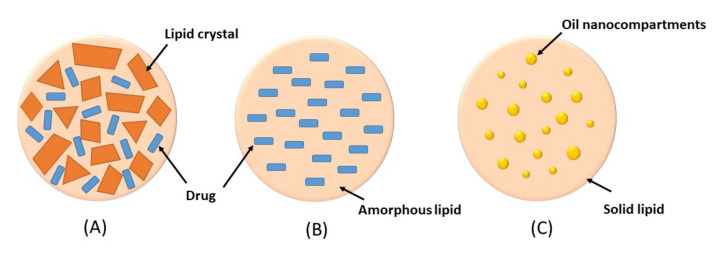
Types of NLCs (**A**) imperfect crystal type, (**B**) amorphous type, (**C**) multiple oil-in-fat-in-water type.

Advantages of NLCs over SLNs [26,27] are:
Higher drug loading capacity by the formation of imperfect matrix and greater drug solubility in liquid lipid in the matrix.NLCs reduce drug expulsion during long-term storage. As known, drug expulsion has been occurred during the ongoing crystallization process of the lipid matrix. By mixing the special liquid oil in NLCs formulation, it forms amorphous structure which limits the crystallization of lipid matrix leading to a reduction of drug expulsion. More flexibility for modulation of drug release by modifying the types and amounts of liquid lipids or surfactants.


Limitations of NLCs [28] are:
Irritation and sensitizing potential of some surfactants. Hwang et al. (2014) revealed that NLC formulation with cationic surfactants induced cell death and the release of inflammatory mediators [29].


## 4. Preparation Methods of SLNs and NLCs

Several methods have been applied for the preparation of SLNs and NLCs, including hot homogenization, cold homogenization, ultrasonication or high shear homogenization, double-emulsion, micro-emulsification, solvent emulsification/evaporation, solvent diffusion and injection, membrane contractor technique, phase inversion technique, and supercritical fluid extraction of emulsions (SFEE) [19].

Hot homogenization

The drug is dissolved or dispersed in the melted lipid. Then, the drug loaded lipid is dispersed in a hot aqueous surfactant phase with continuous stirring by high-shear mixer to make the pre-emulsion. The resulting pre-emulsion is subjected to high-pressure homogenization at a temperature above the lipid melting point to obtain hot oil-in-water (o/w) nanoemulsion. After the homogenization step, the nanoemulsion obtained is solidified by cooling to room temperature to form SLNs. However, high-pressure homogenization can increase the temperature of the mixture resulting in the degradation of heat-sensitive drug.

Cold homogenization

This technique was developed to overcome the problems of hot homogenization, including drug degradation, drug partitioning into the aqueous phase during pre-emulsion step and crystallization of the nanoemulsion which can lead to a low drug loading efficiency [11]. Briefly, the drug is dissolved or dispersed in the melted lipid to make pre-emulsion which is then rapidly cooled by dry ice or liquid nitrogen. The resulting solidified drug-loaded lipid is ground using a powder mill to obtain microparticles. The latter is dispersed in a chilled surfactant solution and homogenized at high-pressure at or below room temperature. The nanoparticles obtained generally have larger size and broader size distribution compared to those prepared by hot homogenization [30].

Ultrasonication or high-shear homogenization

Compared to high-pressure homogenization, this method is more rapid and simple to perform. The drug-loaded melted lipid and an aqueous phase containing surfactant are separately heated at a temperature above the lipid melting point. The aqueous phase is then added to the lipid phase and homogenized using an ultrasonic probe sonicator or high-shear homogenizer. The use of proper amount and type of surfactant will allow the formation of lipid nanoparticles using a simple ultrasonication or high-shear homogenization method [31]. However, metal contamination has to be considered when an ultrasonic probe sonicator is used [30].

Micro-emulsification

This technique is based on the dilution of microemulsions. The drug-loaded melted lipid and an aqueous phase containing surfactant/co-surfactants are separately heated at the temperature above the lipid melting point. Hot o/w microemulsion is produced by adding lipid phase into aqueous phase. The resulting hot microemulsion is then diluted with cold water (2–10 °C) under gentle stirring to form lipid nanoparticles. Typically, the ratio of microemulsion to cold water lies in the range of 1:25 to 1:50. Due to the dilution, the nanoparticles obtained will have lower lipid content than those prepared by the homogenization method [30]. Furthermore, this method is useful for encapsulating drugs which are sensitive to mechanical stress [19,32].

Double-emulsion

This method was developed for the encapsulation of highly hydrophilic drugs in lipid matrix of nanoparticles. In this method, the drug is dissolved in water while the lipid is dissolved in an organic solvent. The water phase is added into the organic phase to form a primary w/o emulsion which is subsequently dispersed in the external water phase to form a double emulsion (w/o/w). The organic solvent is then removed, resulting in the formation of lipid nanoparticles [19].

Solvent emulsification/evaporation

This technique is based on the precipitation of particles in o/w emulsion. First, the lipid and drug are dissolved in water-immiscible organic solvent, then emulsified with an aqueous phase to form an o/w emulsion. The organic solvent is then evaporated leading to the precipitation of lipid in the aqueous medium and formation of nanoparticles. The advantage of this method is the avoidance of heat exposure. Its main disadvantage is the use of organic solvent [30].

Solvent diffusion and injection

The lipid and drug are dissolved in an organic solvent. The resulting solution is then injected through a syringe into an aqueous phase containing surfactant under continuous stirring. Surfactant reduces interfacial tension between water and solvent leading to the formation of small droplets of solvent phase at the injection site. The solvent will then diffuse out of the droplets into the aqueous phase resulting in a droplet size reduction and local supersaturation of lipid within the droplets. Nanoparticle dispersion is formed due to lipid precipitation after solvent evaporation. The advantages of this method are avoidance of heat exposure and small particle size obtained. However, the possibility of having residual organic solvent in the nanoparticles produced is a disadvantage of this method [19].

Membrane contractor technique

The melted lipid phase is pressed through the pores of a membrane at a temperature above the melting point of the lipid to form small droplets which are detached from the membrane pore outlet by the tangential flow of water. SLNs are formed by cooling of the dispersion of droplets to room temperature [33].

Phase inversion technique

This technique is based on the phase inversion temperature (PIT) of some non-ionic polyethoxylated surfactants. The surfactants will form w/o emulsions at temperature above PIT and o/w emulsions at temperature below PIT. In this method, the lipid, drug, and water (containing surfactants) are mixed and heated at a temperature above PIT under constant stirring to form a w/o microemulsion. The microemulsion is then rapidly cooled to a temperature below PIT to form an o/w nanoemulsion, followed by the formation of SLNs at a temperature below the melting point of the lipid [32].

Supercritical fluid extraction of emulsions (SFEE)

SFEE is a novel method for the preparation of SLNs and NLCs. This method uses a supercritical fluid, such as carbon dioxide, to remove the organic solvent from an o/w emulsion, leading to the precipitation of the lipid with the drug as composite particles. Formation of nanoparticles with narrow size distributions can be achieved by this method [6].

## 5. Characterization of SLNs and NLCs

In order to assure the quality and stability of SLNs and NLCs, their properties such as drug loading, entrapment efficiency, particle size and size distribution, zeta potential, degree of crystallinity and co-existence of additional colloidal structures need to be evaluated. Each characterization method mentioned above is summarized below.

Drug loading and entrapment efficiency:

Drug loading (DL) refers to the amount of drug loaded per unit weight of the nanoparticles, indicating the mass of the nanoparticles contributed by the encapsulated drug. Drug loading is calculated using the following equation.
%DL = (Weight of entrapped drug/Weight of nanoparticles) × 100(1)

Encapsulation efficiency is the percentage of drug that is successfully entrapped in the nanoparticles. It can be determined as the ratio between the amount of entrapped drug and the amount of drug added in the preparation of the nanoparticles [18]. EE can also be determined from the amount of drug found in the non-lipid phase (free drug) as shown below:%EE = (Weight of entrapped drug/Weight of drug added) × 100(2)
%EE = [(Weight of drug added − Weight of free drug)/Weight of drug added] × 100(3)

The entrapped drug in particles can be separated from free drug by using various techniques, such as ultracentrifugation. The sedimented particles are collected and dissolved with suitable solvent. The resulting solution is subjected to proper analytical method such as high-performance liquid chromatography (HPLC), spectrophotometrically in order to quantify the entrapped drug. For free drug content, it can be obtained by analyzing the separated supernatant [34].

Particle size and size distribution

Dynamic light scattering (DLS) and laser diffraction are established techniques for the characterization of particle size. These methods can measure particle size across the range of 0.1 nm to 10 µm and 0.01 to 3500 µm, respectively. In DLS, the intensity fluctuations of scatter light from particle motion are measured and used to determine the diffusion coefficient and the particle size. The smaller particles show faster fluctuations of scattered light than larger particles. For laser diffraction, it measures the intensity of light scattered at various angles as a laser beam passes through a dispersion which is then analyzed to calculate the particle size. Large particles scatter light at small angles, whereas small particles scatter light at large angles. Polydispersity index (PDI or PI), a parameter calculated from DLS, is used to indicate particle size distribution. A PDI value of 0.3 and below is considered to be acceptable for representing monodispersity of nanoparticles [30].

Zeta potential:

Zeta potential is the net charge on the particle surface. A greater zeta potential will result in electrostatic repulsion between dispersed particles, thus preventing particle aggregation. Zeta potential can be measured by electrophoretic light scattering. A zeta potential value of ±30 mV is generally considered to result in sufficient electrostatic repulsion force to ensure better physical stability of nanoparticle dispersion [18].

Degree of crystallinity:

The crystallinity behavior of lipid nanoparticles affects the mobility of the entrapped drug. The lower crystallization degree can result in faster drug release due to high mobility of the drug. Differential scanning calorimetry (DSC) and X-ray diffraction (XRD) are commonly used to investigate crystallinity behavior of lipid carriers. In DSC, the heat energy uptake in a sample is compared to a reference and applied to monitor the changes of phase transitions. For XRD, the sample is irradiated with X-rays and the intensity of the radiation scattered at different angles is analyzed. XRD provides structural information such as phases, crystal orientations, crystallinity, and crystal defects of the lipid nanoparticles [30].

Co-existence of different colloidal species:

The co-existence of different colloidal species such as micelles, mixed micelles, and liposomes can solubilize drug, serve as an alternative location for drug incorporation, and affect the stability and release kinetics of the incorporated drug. Therefore, the co-existing colloidal species should be taken into consideration for the development of SLN or NLC formulations and the interpretation of their analytical results. Nuclear magnetic resonance (NMR) is a useful technique to investigate dynamic phenomena and the presence of nano-oil compartments in the colloidal lipid dispersions. NMR can differentiate between different nuclei, elements, and isotopes due to the fact that each specific nuclide will absorb electromagnetic radiation at a very specific frequency [30]. NMR spectra can provide information concerning the number of nuclei and the interaction of nuclei with the surrounding environment. For those who want to get more details regarding the applications of NMR to the study of physicochemical properties of different types of surfactant aggregates, micelles, and related system, many excellent publications should be consulted [35,36].

In vitro drug release:

This characteristic can be measured by dialysis bag diffusion technique. In dialysis technique, lipid dispersion is added into a dialysis bag and then immersed in a dissolution medium at controlled temperature under continuous stirring. The aliquots of the dissolution medium are withdrawn at appropriate time intervals and simultaneously replace with same volume of fresh dissolution medium. The concentration of drug in the aliquots is analyzed by appropriate methods such as UV-Vis spectrophotometer and HPLC [37].

## 6. SLNs and NLCs as Topical Carriers for Anti-Acne Phytochemicals

SLNs and NLCs were broadly studied and used in a wide variety of pharmaceutical and cosmetic applications including anti-acne. Raza et al. (2013) prepared isotretinoin (ITR)-loaded SLNs and NLCs. They evaluated skin transport characteristics and antimicrobial activity against *P. acnes* of these nanoparticles [38,39]. The ITR-loaded SLNs and ITR-loaded NLCs had much lower minimum inhibitory concentration compared to isotretinoin alone. In skin permeation and retention studies, the SLNs and NLCs gave higher permeation flux and skin retention values than those of the marketed product. Liu et al. (2007) reported that SLNs improved skin accumulation of ITR with no penetration through skin, thus avoiding systemic uptake of drug [40]. According to the work of Pokharkar et al. (2014), benzoyl peroxide-loaded SLNs exhibited strong anti-*P. acne* activity with controlled drug release, thereby reducing irritation potential when compared to marketed product [41]. SLNs loaded with triamcinolone acetonide (TA) exhibited prolonged drug release for 24 h. Moreover, the developed TA-loaded SLNs mainly accumulated in the epidermis which might avoid undesirable systemic side effect [42]. Gel containing mometasone furoate-loaded SLN showed significantly superior sustained release and skin deposition than both mometasone furoate-loaded gel (without SLNs) and commercialized product [43]. Pople et al. (2006) reported that SLNs provided prolonged release of vitamin A palmitate up to 24 h, and the SLNs-enriched gel demonstrated a better localization of the drug in the skin than conventional gel [44]. Spironolactone-loaded NLCs in carbopol gels were developed by Kelidari et al. (2016). Their clinical study was performed in patients with mild to moderate acne for 8 weeks. It was found that the NLCs-based gel enabled high tolerance, high skin hydration, and high efficacy against acne compared to spironolactone alcoholic gels [45]. The efficacies and safety of topical anti-acne drugs, such as clindamycin, adapalene, azelaic acid, isotretinoin, salicylic acid, and retinyl palmitate, were improved by incorporating these drugs into SLNs and NLCs [46,47,48,49,50,51]. The encapsulation of various anti-acne phytochemicals in SLNs and NLCs was also studied by several research groups as described in the following sections.

### 6.1. SLNs as a Promising Carrier System for the Topical Delivery of Anti-Acne Phytochemicals

SLNs loaded with resveratrol, vitamin E, and epigallocatechin gallate (EGCG) for skincare applications have been developed by Chen et al. in 2017. The study showed that lipid nanoparticles provided protective effect against UV-induced degradation of resveratrol and vitamin E and improved skin penetration of resveratrol [52]. Previous studies showed that the topical formulations containing resveratrol and EGCG were effective in reducing inflammation, sebum production, and the viability of *P. acnes* [53] as well as reducing the severity of *acne vulgaris* in patients [54]. Shrotriya et al. (2018) developed SLNs gel loaded with curcumin with the aim of improving its efficacy. Curcumin is a phytochemical extracted from the rhizome of *Curcuma longa* (Zingiberaceae family). Curcumin has anti-inflammatory and antimicrobial activities which may combat the bacteria that contribute to acne. The results demonstrated that SLNs-based gel gave better occlusive effects and skin accumulation of curcumin compared to plain gel. The optimized curcumin-loaded SLNs had mean particle size of 51 nm and entrapment efficiency of 93% [55]. According to Kakkar et al. (2018), SLNs loaded with tetrahydrocurcumin (THC), a partially reduced derivative of curcumin, provided great occlusive effect. The THC- loaded SLNs in gel formulation demonstrated better therapeutic effects than free THC [56]. It was also found that the formulation containing just 10% SLNs resulted in better occlusion properties than the gold standard (Vaseline) [57]. Talarico et al. (2021) reported that the controlled release of Quercetin, a poorly water-soluble flavonoid, over 26 h was achieved with SLNs composed of stearic acid as core lipid and Arabic Gum as stabilizer. In addition, the SLNs were found to enhance antioxidant activity compared to free Quercetin [58]. Eugenol is a natural compound widely found in many aromatic plant species such as clove, holy basils, and betel vine. It has shown anti-acne activity by suppressing *P. acnes*-induced inflammatory reaction [59]. Garg et al. (2014) studied the permeation of hydrogel containing eugenol-loaded SLNs in the epidermis of human cadaver skin. It was found that the SLNs could enhance the accumulation of eugenol in the epidermis compared to plain hydrogel. Eugenol-loaded SLNs also exerted greater occlusion and hydration to the skin than eugenol oil and plain hydrogel [60]. Resveratrol (3,5,4′-trihydroxystilbene) is a polyphenolic compound found naturally in grapes, wine, peanuts, cocoa, and some berries and it has multiple biological effects including antimicrobial properties against *P. acne* [61]. According to Rigon et al. (2016), resveratrol (RES)-loaded SLNs could be used for administration of RES to improve its efficacy in skin disorders such as hyperpigmentation and aging [62].

### 6.2. NLCs as a Promising Carrier System for the Topical Delivery of Anti-Acne Phytochemicals

Rapalli et al. (2020) developed curcumin-loaded NLCs with the aim of improving its skin permeability. The results indicated that the skin permeation of curcumin-loaded NLCs was three times higher than that of curcumin alone. Moreover, curcumin loaded-NLCs showed extended in vitro release up to 48 h [63]. Lacatusu et al. (2017) studied the anti-inflammatory activity of the marigold extract and azelaic acid co-loaded NLCs. The results showed that the NLCs could reduce inflammatory IL-6 and IL-1β cytokines tested by ELISA method and paw edema in rats challenged with carrageenan [64]. Moreover, a synergistic effect of carrot extract (CE) combined with azelaic acid (AA) in NLCs on anti-inflammatory and anti-acne activities was observed by Lacatusu et al. (2020). The results revealed that the NLCs exerted superior anti-inflammatory effect compared with the commercial product. Furthermore, the expression of inflammatory IL-1β and TNF-α cytokines was decreased in the cells treated with CE-AA loaded NLCs [65,66]. Salicin is an alcoholic β-glucoside found in willow bark extract which is used to treat skin diseases such as acne due to its anti-inflammatory and high comedolytic activities. According to Arsenie et al. (2020), NLCs loaded with a mixture of white willow bark extract (WBE), azelaic acid and panthenol were able to improve the epidermal cell reconstruction. The gel containing the NLCs gave a degree of hydration of 84 % in the T-zone for type-III skin (predisposed to acne) [67]. Asiaticoside, madecasosside, asiatic acid, and madecassic acid are the phytochemicals found in *Centella asiatica* which exhibit multi-therapeutic effects including antioxidant, anti-inflammatory, antimicrobial, and anticarcinogenic activities [68]. NLCs were found to enhance the membrane fluidity of stratum corneum which enabled the asiaticoside in *Centella asiatica* to penetrate the skin [69]. Singh et al. (2016) studied the anti-inflammatory effect of silymarin- loaded NLCs. Silymarin is a phytochemical extracted from *Silybum marianum* (milk thistle) fruit. It has comedolytic activity and reduces the production of fatty acid [70,71]. It was found that silymarin-loaded NLCs could reduce the swelling and inflammation of mouse skin induced by 7,12-dimethylbenz[a]anthracene (DMBA). The NLCs also improved the stability of silymarin [72]. Rosli et al. (2015) reported that NLCs loaded with *Zingiber zerumbet* oil, a phytochemical found in *Zingiber zerumbet* (pinecone or shampoo ginger), were stable and suitable for transdermal delivery systems [73]. *Zingiber zerumbet* oil itself was found to have anti-*P. acnes* activity (MIC ≥ 125 μg/mL), great antioxidant and anti-biofilm formation activities [74]. *Origanum vulgare* L. or oregano is an aromatic plant containing high amounts of carvacrol and thymol essential oil. It possesses antimicrobial activity against *P. acnes* with MIC of 0.34 mg/mL [7,18]. According to Carbone et al. (2018), NLCs loaded with this essential oil exhibited antioxidant and anti-inflammatory effects and promising drug solubility and stability enhancement properties [75]. Bose et al. (2021) studied the effect of α-terpineol- loaded NLCs against *Pseudomonas aeruginosa* in the murine model. Alpha-terpineol is a phytochemical found in tea tree oil that strongly inhibits *P. acnes* with MIC of 2.5%. It was found that topical application of α-terpineol-loaded NLCs could reduce bacterial count in the infected tissue and decrease the levels of various inflammatory markers [39,76]. Cannabidiol (CBD) is a non-psychoactive phytocannabinoid found in *Cannabis sativa*. CBD is one of the promising anti-acne phytochemicals due to its anti-inflammatory and sebostatic activities [30]. Esposito et al. (2016) developed and optimized the formulation of CBD-loaded NLCs prepared by ultrasonication method and initiated its clinical trial [77]. Cinnamon oil is an essential oil obtained from *Cinnamomum zeylanicum* or cinnamon bark. The main components of cinnamon oil are cinnamaldehyde and eugenol [23]. Cinnamon oil exerts anti-*P. acnes* activity with MIC of 5 mg/mL. Wen et al. (2018) developed cinnamon-loaded NLCs for promoting wound healing. The study showed that NLCs containing cinnamon oil had antimicrobial activities against antibiotic-resistant *P. aeruginosa* and *Burkholderia cepacia* complex. The NLCs also enhanced skin permeation of cinnamon oil and could shorten the treatment time [78]. Essential oil from bergamot or *Citrus medica* L. var. *sarcodactyl* has shown anti-acne activity by promoting apoptosis of sebaceous glands and inhibiting the accumulation of triglycerides and inflammation [79]. Bergamot oil-loaded NLCs were developed and studied for the photodynamic treatment of vitiligo. Research findings suggested that bergamot oil was successfully loaded into NLCs, which provided sustained release of bergamot oil for 24 h [80].

In addition to the above information, brief details of various formulations and their respective findings are mentioned in Table 1.

For this mini-review, the references cited are more specific to the application of SLNs and NLCs to topical delivery of phytochemicals for the treatment of *acne vulgaris*. Readers can however refer to the literature to find out more about the materials and methods used for the production of SLNs and NLCs, as well as their other applications in topical, dermal, and transdermal drug delivery. Such information can be obtained from many insightful review articles previously published [6,16,89,90,91,92,93,94,95,96,97,98].

## 7. Conclusions

The SLNs and NLCs are attractive and promising lipid nanocarriers for topical delivery of phytochemicals due to their desirable properties that include skin penetration enhancement, promising occlusive effect, possibility to modulate drug release kinetics, ability to prevent the degradation of phytochemicals and suitability as carriers for both hydrophilic and lipophilic active substances. Moreover, the SLNs and NLCs can be applied onto damaged or inflamed skin because they are composed of biocompatible and non-toxic lipids. However, it is worth highlighting that although remarkable results of SLNs and NLCs as delivery systems for anti-acne phytochemicals have been demonstrated by many research groups, their in vivo efficacy in treating acne has not been fully established yet. Hence, further investigation on the potential of these lipid carriers in clinical setting is highly warranted and strongly encouraged. This would bring a new perspective on SLNs and NLCs as phytochemical carriers for topical treatment of acne.

## Figures and Tables

**Table 1 molecules-27-03460-t001:** Examples of SLNs and NLCs loaded with anti-acne phytochemicals.

Lipid Carriers	Phytochemicals	Compositions	Findings	References
SLNs	Quercetin	Tristearin, phosphatidylcholine	The penetration through the stratum corneum of quercetin via quercetin-loaded SLNs in topical emulsion (21.2 ± 2.9%) was greater than that of control emulsion (18.1 ± 2.3%).	[81]
SLNs	Curcuminoids	Beeswax, Tween 80, lecithin	The formulation showed a sustained release with first order kinetics.	[82]
SLNs	Neem oil	Soya lecithin, cholesterol, Tween80	Entrapment efficiency (EE) of neem oil-loaded SLNs was in the range of 67.23–82.10%. EE increased when the concentrations of lecithin and tween 80 increased. SLNs showed burst release (3.56–30.05%) within first 30 min.	[83]
SLNs	Resveratrol	Glyceryl behenate (Compritol 888), poloxamer 188 (Pluronic F68), Tween 80, Miglyol^®^ 812	Mean particle size of SLNs and NLCs were 287.2 nm ± 5.1 and 110.5 nm ± 1.3, respectively. The drug entrapment efficiency was 18% higher in NLCs. NLCs penetrated deeper into the skin than SLNs.	[84]
SLNs	Propolis flavonoids (PFs)	Glyceryl monostearate, soy lecithin, PEG400, Tween80	The PFs-loaded SLNs exhibited prolonged drug release for 24 h and prolonged anti-inflammatory properties. No cytotoxicity observed.	[85]
NLCs	Resveratrol	Cetyl palmitate, sesame oil, tween 80 Glyceryl behenate (Compritol 888), sesame oil, tween 80 Phospholipid (Phospholipon80), sesame oil, Tween 80	All NLC formulations showed a slow degradation of resveratrol over 24 h while resveratrol solution showed rapid degradation in the first 8 h. Phospholipon-based NLCs showed sustained release of resveratrol over 24 h and improved the penetration of resveratrol through stratum corneum.	[52]
NLCs	α-Mangostin	Cetyl palmitate, lavender oil, Montanov 82, Polyoxyethylene (20) sorbitanmonolaurate, poloxamer and glycerol	α -Mangostin-loaded NLC reduced the levels of inflammatory mediators including nitric oxide and TNF-α in macrophages induced by lipopolysaccharide.	[86]
NLCs	Eucalyptus essential oils or Rosemaryessential oils	Cocoa butter, olive oil or sesame oil, lecithin	All NLC formulations showed good bioadhesive properties. Eucalyptus oil-loaded NLCs prepared with olive oil showed antimicrobial and wound healing properties.	[87]
NLCs	Lycopene	Eumulgin SG, orange wax, rice oil	Lycopene-loaded NLCs gave a biphasic release profile (a relatively fast release during the first 6 h, followed by a sustained release during the next 18 h). The occlusive properties of NLC increased with increasing lycopene loading. The utilization of NLC increased the stability of lycopene.	[88]

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
