# Peer review of "A Mini-Review on Solid Lipid Nanoparticles and Nanostructured Lipid Carriers: Topical Delivery of Phytochemicals for the Treatment of Acne Vulgaris"

_molecules, 2022, doi:10.3390/molecules27113460_

Round 1

Reviewer 1 Report

The mini-review is interesting and I consider it would be well-accepted by expert readers. It requires some minor changes as shown in the attached pdf file. In particular the following:

  1. All the Latin words as scientific names must be written with italics.
  2. The chapter about lipid particles applications could be moved to the end after particles characterization.

Reviewer 2 Report

The review work entitled: "A mini-review on solid lipid nanoparticles and nanostructured  lipid carriers: Topical delivery of phytochemicals for the treatment of acne vulgaris " presents a content deficiency; there wasn't an in-depth study on the theme proposed.Several review articles published previously and with superior quality it could be mentioned. In addition, the authors should have delved into the topic and presented more consistent scientific studies that could encourage researchers in the area to search for their review. Finally, less than 70 references are mentioned for a literature review work in this work, as a clear demontration of weak review paper. I suggest rejecting this submission at the moment.

Reviewer 3 Report

The manuscript “A mini-review on solid lipid nanoparticles and nanostructured lipid carriers: Topical delivery of phytochemicals for the treatment of acne vulgaris” gave a short perspective of the solid lipid nanoparticles (SLNs) and nanostructured lipid carriers (NLCs) as promising delivery systems for the treatment of acne vulgaris. Before publication, personally, I think a few things are needed to be addressed.

First, Line 131 and Line 173, “irritation and sensitizing potential of some surfactants” needs to be specified. Since both have been shown in SLN and NLC as limitations.

Figure 2(A), the label for lipid crystal and drug are unclear.

For section 4, “SLNs and NLCs as topical carriers for anti-acne phytochemicals”, the author used different drugs or effective chemicals as subtitles. However, I think it will be better to rearrange this section, using different SLNs and NLCs as the subtitles. The reason is this review is for SLNs and NLCs using in treating acne vulgaris, not different drugs for acne vulgaris. It will be better to highlight each different nanoparticle system than drugs. Then it will make sense to show table 1, which summarized everything, including different drug kinds.

For section 5 preparation and section 6 characterization, I personally will suggest to put them before the application (section 4). Normally, people talked about preparation, characterization and then applications.

Of course, these are my personally suggestions, if the author doesn’t agree, they can be addressed and explained too.

Round 2

Reviewer 3 Report

No more comments after this revision.